# Coherent Pulse-Compression Lidar Based on 90-Degree Optical Hybrid

**DOI:** 10.3390/s19204570

**Published:** 2019-10-21

**Authors:** Jing Yang, Bin Zhao, Bo Liu

**Affiliations:** 1University of Chinese Academy of Sciences, Beijing 100049, China; yangjing16@mails.ucas.ac.cn; 2Institute of Optics and Electronics, Chinese Academy of Sciences, Chengdu 610209, China; zhaobin@ioe.ac.cn

**Keywords:** chirp pulse compression, coherent lidar, hybrid

## Abstract

A coherent pulse-compression lidar system based on a 90-degree optical hybrid is demonstrated in this paper. In amplitude modulation (AM) mode, the returned RF chirp signal will be influenced by a random phase difference between local oscillator and echo light, causing fluctuations in the ranging results, and as a result the detection probability is small. By using the 90-degree optical hybrid, two orthogonal complementary signals are obtained to stabilize the result so as to increase the detection probability. We performed an experiment to measure the distance of a white printed wall which is about 65 m away from the system. The detection probability increased from 65% to 99.88%, and the precision is improved from 0.42 m to 0.27 m.

## 1. Introduction

Pulse-compression technology is used to compress a long pulse chirp signal into a narrow pulse in time by convolution [1]. Since the pulse is greatly compressed, the energy is concentrated in a narrow pulse, so the signal-to-noise ratio (SNR) and resolution are increased significantly. Therefore, this system only requires the transmitting peak power of mW magnitude.

It is worth noting that some pulse-compression systems are exactly the same as standard frequency modulated continuous wave (FMCW) scheme [2,3,4], except that their signals are pulsed. Both of them obtain the target distance through beat frequency between local oscillator and echo light, so they can only utilize the overlap part in one sweep period. The farther the target is, the less overlap there will be. But our pulse-compression scheme has no such limitations, because it compresses a long pulse into a much narrower pulse whose peak position indicates the target distance through matched filter, namely local chirp to convolving with signal, therefore it is more advantageous for long-distance detection.

The compression ratio is equal to the time bandwidth product (TBP) of the chirp pulse. Some chirp pulse-compression lidar systems chirp laser through frequency modulation (FM) [4,5,6,7,8,9,10,11]. But each FM system has its own limitation: In ref [4], the system uses the acoustic optical modulator (AOM) to chirp laser. Since its bandwidth is only 14 MHz, the TBP is small, so the resolution is limited; In refs [5,6], these systems use a ceramic piezoelectric transducer (PZT) to modulate the length of laser cavity to chirp laser. PZT has the drawback of instability and nonlinearity of modulation, so the ranging results will fluctuate and the SNR is small. In refs [7,8], these systems use dispersive element to chirp laser, which is only suitable for fiber sensing because free space will greatly deteriorate the linearity of the chirp optical signal; Other systems are either complicated or bulky [9,10,11]. In contrast, AM system based on electro-optic modulator (EOM) has great agility, such as large bandwidth and fast response speed.

Our system based on EOM has a small size, simple structure, and is suitable for measuring long-distance hard target in free space. In our system, there is a random phase difference between the echo and local oscillator light, which is caused by atmospheric turbulence, laser phase noise, and target vibration, so it is a random variable. The random phase difference makes the amplitude of the detected electrical chirp signal randomly fluctuate, sometimes smaller than noise even after compression, so the detection probability is small. Detection probability means the probability we determine that we detect a target under a certain SNR when a target does exist. In order to solve this problem, we use a 90-degree optical hybrid at the receiver [12]. Connecting the hybrid with two balanced detectors, we obtain two orthogonal electrical signals, I and Q—if one signal is weakened, the other one will be enhanced, so we can use them to stabilize the ranging results.

In addition, comparing with other coding ranging system [13], our system can also be used to detect the radial velocity and distance of the target easily at the same time without any change, which has been verified in our previous work [14].

## 2. Materials and Methods

### 2.1. Principle

The principle of the system is as shown in Figure 1.

Figure 1 shows the block diagram of the proposed lidar system. It is worth noting that all devices are polarization–maintaining. A coupler divides the power of the 1550 nm laser (Pure Photonics PPCL550) into 90% and 10% part. The 90% part is modulated by an EOM (IXblue MXAN-LN-10) with an RF chirp pulse produced by a signal generator (Tektronics AFG3000). The modulated light is injected into port 1 of the optical circulator and then injected into the telescope through port 2. The echo light is obtained by the same telescope and sent to the hybrid (kylia COH24) through port 3. The circulator is not a perfect instrument so parts of the modulated light will leak from port 1 to 3 directly, namely crosstalk. This is useful and important because the crosstalk can be used as a zero-time base. The 10% part is used as local oscillator light. It mixes with echo and crosstalk in the hybrid. The mixed light is detected by two balanced detectors (THORLABS PDB460C-AC) [15,16]. The detected RF pulse convolves with local RF chirp wave to perform the compression so as to get the distance of the target.

For the convenience of derivation, first, we only analysis the echo light. The chirp voltage waveform used to drive the modulator is
(1)$$V_{s}(t)=V_{D}\cos(2\pi f_{1}t+\pi \frac{f_{2}-f_{1}}{\tau }t^{2})=V_{D}*\cos[V(t)]$$
where $V_{D}$ is the amplitude of the driving voltage signal, τ is the pulse duration. When the modulator is based at a null point, the output optical field of the modulator is
(2)$$E_{o}=E_{s}\sin(\frac{\pi V_{s}(t)}{V_{\pi }})\cos(2\pi f_{0}t)$$
where $E_{s}$ is the amplitude of optical field, $V_{\pi }$ is the half wave voltage of EOM, $f_{0}$ is the center frequency of laser source. For small-signal modulation, Equation (2) can be approximated as
(3)$$E_{o}\approx m\cos[V(t)]\cos(2\pi f_{0}t),\;m=E_{s}\pi V_{D}/V_{\pi }$$

When the target is stationary, the echo and local optical fields are
(4)$$E_{R}=a\cos[V(t^{\prime })]\cos(2\pi f_{0}t^{\prime }),\;E_{L}=b\cos[2\pi f_{0}t]$$
where *a* and *b* are the amplitude of them and $t^{\prime }$ denotes the time of echo to reach receiver. So the optical field E1 to E4 exiting the four output ports are

(5)$$\left(\begin{array}{l} E_{1}\\ E_{2}\\ E_{3}\\ E_{4} \end{array}\right)=\frac{1}{2}\left(\begin{array}{cc} \begin{array}{l} 1\\ \exp(j\frac{\pi }{2})\\ \exp(j\frac{\pi }{2})\\ \exp(j\pi ) \end{array} & \begin{array}{l} \exp(j\frac{\pi }{2})\\ 1\\ \exp(j\frac{\pi }{2})\\ 1 \end{array} \end{array}\right)\left(\begin{array}{c} E_{R}\\ E_{L} \end{array}\right)$$

The corresponding photocurrents are
(6)$$\begin{array}{l} I_{arm1}=R|E_{1}{|}^{2}=R\{i_{D}+\frac{ab}{2}\cos[V(t^{\prime })]\cos(\theta -\frac{\pi }{2})\}\\ I_{arm2}=R|E_{2}{|}^{2}=R\{i_{D}+\frac{ab}{2}\cos[V(t^{\prime })]\cos(\theta +\frac{\pi }{2})\}\\ I_{arm3}=R|E_{3}{|}^{2}=R\{i_{D}+\frac{ab}{2}\cos[V(t^{\prime })]\cos(\theta )\}\\ I_{arm4}=R|E_{4}{|}^{2}=R\{i_{D}+\frac{ab}{2}\cos[V(t^{\prime })]\cos(\theta -\pi )\} \end{array}$$
(7)$$i_{D}=\frac{a^{2}}{8}{\left[V(t^{\prime })\right]}^{2}+\frac{b^{2}}{4},\theta =f_{0}(t^{\prime }-t)=f_{0}*\Delta t$$
where *R* is the responsivity of the photodetectors. The Δt is a time delay, and *θ* denotes a phase difference between echo and local oscillator light. The output voltage of the balanced detector is proportional to the difference between its two photocurrents [9,10]. Then the output voltages of the two balance detectors are
(8)$$\begin{array}{l} V_{I}=Gab\cos[V(t^{\prime })]\sin(\theta )\\ V_{Q}=Gab\cos[V(t^{\prime })]\cos(\theta ) \end{array}$$
which are the coherent terms of echo with local oscillator. Next, we include the crosstalk. Since the photodetection is a nonlinear process, crosstalk, echo and local oscillator light will influence each other, namely, there will be three coherent terms. However, except for echo, the crosstalk is also at least four orders of magnitude smaller than the local oscillator light, so the coherent term of echo and crosstalk can be ignored, leaving only these two coherent terms: crosstalk with local, echo with local. Then, after considering the crosstalk, the detected electrical signals are

(9)$$\begin{array}{l} V_{I}=Gab\cos[V(t^{\prime })]\sin(\theta )+Ga^{\prime }b\cos[V(t)]\sin(\alpha )\\ V_{Q}=Gab\cos[V(t^{\prime })]\cos(\theta )+Ga^{\prime }b\cos[V(t)]\cos(\alpha ) \end{array}$$

Among them, $a^{\prime }$ denotes the amplitude of the crosstalk and α denotes the phase difference between crosstalk and local. After convolutional compression, we can get two narrow pulse envelopes of crosstalk and echo respectively. Then the distance d of the target can be obtained from the difference in time between crosstalk and echo

(10)$$d=c\left(t^{\prime }-t\right)/2=c\Delta t/2$$

### 2.2. System Description

An experiment was conducted to demonstrate the improvement of the 90-degree optical hybrid in detection probability. The center frequency of the laser source was set at 193.4 THz and the power was set at 50 mW. The frequency of the RF chirp signal sweeps from 1 MHz to 99 MHz, the pulse duration is 10 μs and the repetition period is 50 μs. Due to insertion loss, the output peak power of the EOM is just 14 mW.

The modulated signal light from the telescope was focused onto the white painted wall at the end of the corridor through free space, as shown in Figure 2.

## 3. Results

A digital oscilloscope (TELEDYNE LECROY Waverunner 610ZI) samples the two channel electrical signals of balance detectors. Then the digital signals were sent to the computer. After data processing through, the distance can be obtained in real time. We sampled and processed the detected electrical signals for 800 times. Three of them are shown in Figure 3.

The chirp pulse consists of crosstalk and echo. Since the time delay between echo and crosstalk is very short compared to the pulse duration, the crosstalk and echo almost overlap in time. In addition, the echo is smaller than the crosstalk that the echo cannot be seen by the naked eye. So, when we explore the trend, we can treat the above signal as crosstalk, and the echo contained in it can be ignored. We can see that I and Q signals are complementary, when one is weakened the other is enhanced. The amplitude of 200 I and Q signals are shown in Figure 4.

In order to more clearly show the complementary characteristics of I and Q, we only show the amplitude of 200 pairs of signals. We can see that their trend is completely in keeping with the law described in Equation (9) and the phase difference θ changes irregularly, so it is a random variable and varies sharply between two consecutive samples, but it can be seen from Figure 3 that within one sample, it has not changed much. So, within one pulse, the θ can be considered unchanged.

One of the data processing method for I and Q channels are as follows: first sum the squares of I and Q, then compress the sum signal, and then make Hilbert transform to the compressed signal to get the envelope. But this method has limitations which will be demonstrated by following analysis.

Although the random phase difference can be eliminated theoretically by adding the squares of I and Q channels, in fact, since crosstalk and echo generated almost overlap in time, the sum of the squares of the two signals is

(11)$$\begin{array}{l} V_{I}{}^{2}+V_{Q}{}^{2}\\ ={\left\{Gab\cos[V(t^{\prime })]\sin(\theta )+Ga^{\prime }b\cos[V(t)]\sin(\alpha )\right\}}^{2}+\{Gab\cos[V(t^{\prime })]\cos(\theta )+\\ Ga^{\prime }b\cos[V(t)]\cos(\alpha ){\}}^{2} \end{array}$$

(12)$$\begin{array}{l} =\frac{Ga^{2}b^{2}}{2}\cos[2V(t^{\prime })]+\frac{G{a^{\prime }}^{2}b^{2}}{2}\cos[2V(t)]+G^{2}aa^{\prime }b^{2}\cos[V(t^{\prime })]\cos[V(t)]\cos(\theta -\alpha )\\ +\frac{G^{2}b^{2}}{2}(a^{2}+{a^{\prime }}^{2})+G^{2}aa^{\prime }b^{2}\cos(\theta -\alpha ) \end{array}$$

In Equation (12), the first term is echo, the second term is crosstalk, the third term is coherent sum of them, and the rest are direct current (DC) term and random noise term. Only the first and third terms contain $t^{\prime }$, so only they can be used to measure the distance of the wall.

Because the sweeping rate of the first term is twice of that of third term, and one local chirp can only match one term, we can only utilize one of them to measure the distance. Firstly, since the local oscillator light is much larger than the crosstalk and the crosstalk is much larger than the echo, namely $a\ll a^{\prime }\ll b$, if we use first term, the other much bigger term will be a non-negligible noise. Secondly, since the third term contains the multiplier: cos(θ−α) in Equation (12), and we have
(13)0≤|cos(θ-α)|≤1
so the amplitude of the third term varies randomly from zero to its maximum, the detection probability cannot be 100% because there is always system noise. After normalization, when the amplitude of noise is 80% of the sum envelope and if the SNR is bigger than 3 dB, we determined that the target is detected, the detection probability is easily calculated as

(14)$$\frac{1-1\times 0.8}{1}=20\%$$

It can be seen from above analysis that the squaring method has its shortcomings, therefore we try another method: first compress I and Q, and then sum their envelope. The data processing steps are as follows: first, the two signals are respectively compressed, and then we make Hilbert transform to each compressed signal to obtain respective envelopes, and then sum the two envelopes. In this method, since there is no nonlinear process of squaring, there are only signal and crosstalk and no other coherent terms like Equation (12) act as noise so the SNR will be bigger. Besides, the detection probability will be improved because
(15)$$1\leq \left|\sin(\theta )\right|+\left|\cos(\theta )\right|\leq \sqrt{2}$$
which means the amplitude of the sum envelope will vary randomly from its minimum instead of zero to maximum. After normalization, when the noise is small than 1, the detection probability is 100%. When the amplitude of noise is 80% of the sum signal, the detection probability is

(16)$$\frac{\sqrt{2}-\sqrt{2}\times 0.8}{\sqrt{2}-1}\approx 68.3\%$$

Which is bigger than 20% as calculated in Equation (14).

In order to reduce the side lobes, we multiplied local chirp with a Hamming window, at the expense of broadening the main lobe. The compressed envelopes of I, Q, and sum in Figure 3 are as shown in Figure 5.

As can be seen from Figure 5, the crosstalk and echo that almost overlap before compression are completely separated after compression. It can be seen from both crosstalk and echo that when one channel is weak, the other is strong, so that the amplitude of sum envelope can always be greater than the noise. It is worth noting that the random phase difference between crosstalk and echo do not change synchronously because they walk different paths, thereby, when one channel of crosstalk is strong, the echo is not likely to be strong. From the crosstalk, we can see that the sum envelope is still unstable, as is described by Equation (15).

By compression, the magnitude of the crosstalk increases from about 0.04 V to about 208 V and the echo signal that is invisible to the naked eye increases to about 17 V. The data of 208 V and 17 V are obtained by averaging the results of 800 compressions.

In order to illustrate the improvement of 90-degree optical hybrid in ranging performance, we use 800 sets of I and Q signals to separately measure the distance and compare their results with using the sum. We only keep those measurement results whose SNR are greater than or equal to 13 dB. The results are shown in Figure 6. The blue points are the valid distance measurement results, and the green line is the mean of the distance.

## 4. Discussion

The detection probability increases by about 35% by using the 90-degree optical hybrid. It can be seen from the Figure 6 that when measuring with one channel alone, due to the random phase difference, the signal amplitude may be so small that the SNR after compression is still lower than 13 dB, so the detection probability is no bigger than 70%. When measuring with sum signal, it is greatly improved by up to 99.88%. The reason why the detection probability of the latter method is not 100% is that sometimes the atmospheric turbulence is so strong that the echo is not almost coherent with the local light, causing the detected two radio frequency (RF) echo chirps to both be so weak that the SNR is less than 13 dB. Still, this is a low probability event since it only happens once in 800 results. We can also see that when measured with the sum signal, the distribution of the blue points is more compact, and the standard deviation reduces from about 0.42 m to about 0.27 m. So, the precision, which means the reproducibility between the results obtained by repeated measurements using the same system set-up, is improved.

Limitation factors of the system ranging precision mainly are: TBP and sampling frequency. Chirp signal of larger TBP and sampling instrument of higher sampling frequency can contribute to higher precision.

We can use the scheme of transmitting and receiving with different telescopes to avoid the crosstalk effect of the circulator, so that the squaring method can be used to make the signal completely stable. However, this scheme needs to match the field of view of the two telescopes. The receiving field of view is typically larger than the transmitting field of view, therefore more noise is introduced. More notably, there is no circulator to act as a zero-time base, therefore we need an additional one, which makes the system more complicated.

## 5. Conclusions

In this paper, we have demonstrated a coherent pulse-compression lidar system based on 90-degree optical hybrid. Experiments show that after using the 90-degree optical hybrid, the detection probability increases from about 65% to 99.88%, and the precision increases from 0.42 m to 0.27 m. In the future, we will optimize this system to enable it to measure farther targets, so it can be used to measure the distance and radial velocity of long-distance hard targets in free space, such as various buildings and cars in cities, while maintaining the advantages of low transmission peak power.

## Figures and Tables

**Figure 1 sensors-19-04570-f001:**
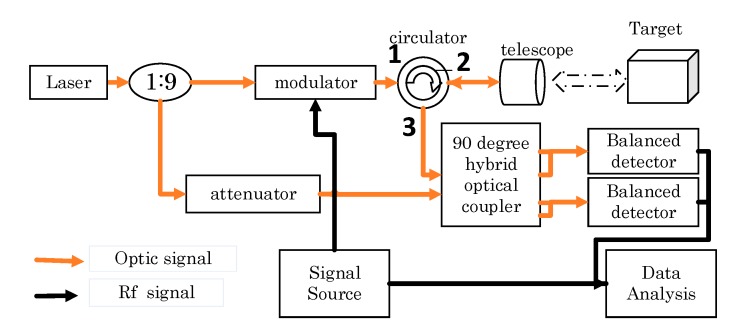
The block diagram of the proposed lidar system. The yellow line denotes the optical path and the black line denotes the electrical path.

**Figure 2 sensors-19-04570-f002:**
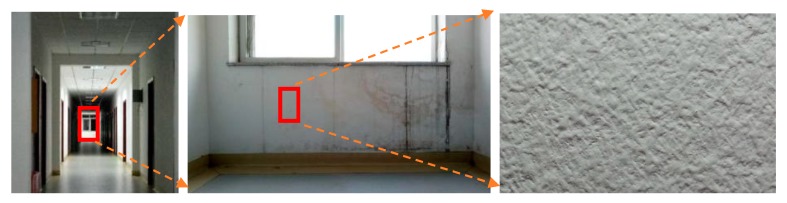
The white painted wall at the end of the corridor works as target.

**Figure 3 sensors-19-04570-f003:**
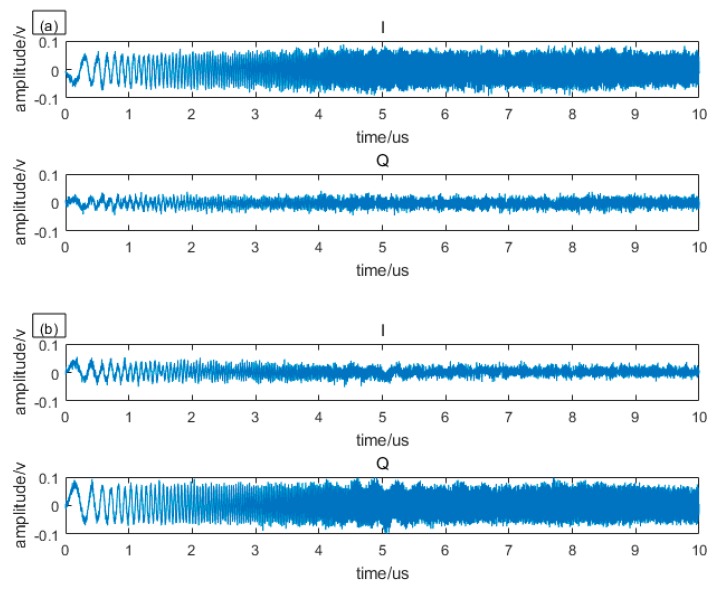
Three cases of detected RF echo are listed as: (**a**) I is strong while Q is weak; (**b**) I is weak while Q is strong. (**c**) I and Q are almost equal.

**Figure 4 sensors-19-04570-f004:**
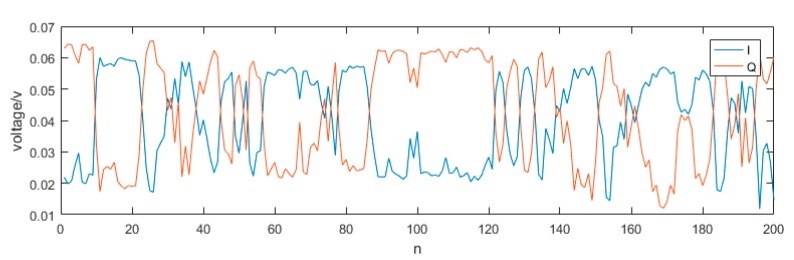
The reflected RF signal collected by the oscilloscope.

**Figure 5 sensors-19-04570-f005:**
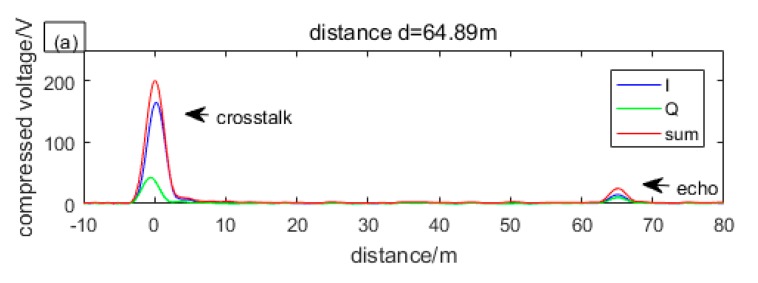
The compressed results of Figure 3 are listed as: (**a**) I is strong while Q is weak; (**b**) I is weak while Q is strong; (**c**) I and Q are almost equal.

**Figure 6 sensors-19-04570-f006:**
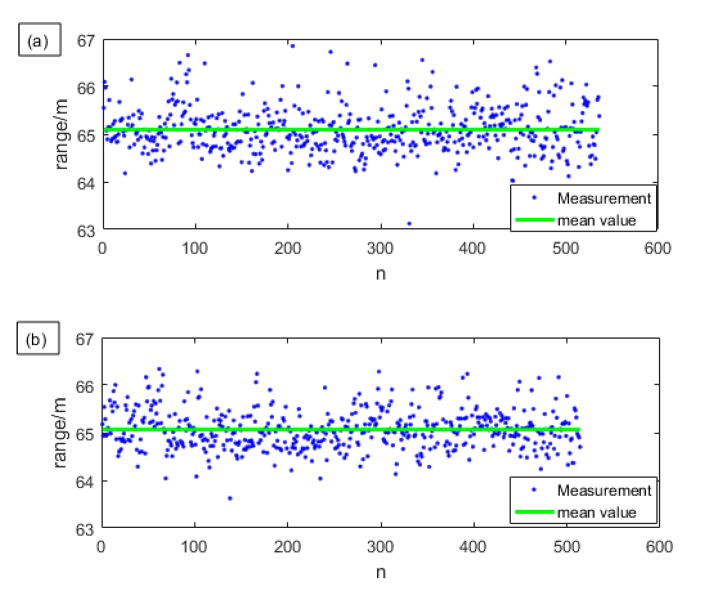
The measurement results are listed as: (**a**) I signal. Number of valid results: 536, average value: 65.1 m. Standard deviation: 0.48 m, probability of detection: 67%; (**b**) Q signal. Number of valid results: 514, average value: 65.07 m. Standard deviation: 0.42 m, probability of detection: 64.25%; (**c**) Sum signal. Number of valid results: 799, average value: 65.00 m, standard deviation: 0.27 m, probability of detection: 99.88%.

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
