# Peer review of "Coherent Pulse-Compression Lidar Based on 90-Degree Optical Hybrid"

_sensors, 2019, doi:10.3390/s19204570_

Round 1
Reviewer 1 Report
A long distance measurement strategy was proposed with high detection probability and relative accuracy as declared. The following points need discussing with the authors.
1.The authors introduced that “In addition, comparing with other coding ranging system [13], our system can also be used to detect the radial velocity and distance of target easily at the same time without any change, which has been verified in our previous work [14]”. This means that the proposed method or system has been published already. What is the novelty in this manuscript?
2.The term “detection probability” is not explained clearly.
3.Fig.2 is not clear. What is the details of the white painted wall?
4.The language should be improved.
Reviewer 2 Report
The paper deals with the description and preliminary experimental characterization of a range finding lidar set-up based on the combination of coherent detection, pulse compression, and a 90° hybrid optical coupler.
Coherent detection and pulse compression are expected, in future experiment, to provide velocity measurement on top of object distance measurement. The main focus of the paper is to characterize how the implementation of the 90° optical hybrid coupler can increase detection and range finding probability.
The paper is well presented and documented. As exposed in the introduction, the combination proposed has some novelty and merit compared with other set-up from the literature and the state of art. The experimental data support the conclusions, and the issue of cross-talk is well explained and tackled. I would consequently give a favorable recommendation for publication in Sensors, provided some revisions that need to be implemented before publication.
Comments questions and recommendations
The overall paper needs to be carefully read and reviewed in order to suppress the remaining incorrect or ambiguous English phrasing as well as the typos. Here are some examples (I probably missed many others) : Line 25 “through beat frequency” instead of “though” Line 27 should be written like : “the farther the target, the less usefull the overlap part will be”. On top of that I would not say the overlap is “less usefull”, it is simply reduced. Fig 1. There is a typo “balanced detector” instead of "detecotor Line 71: “for the convenience of deviation” I do not understand what the authors mean? Line 151 : “the sweeping rate of the first term is twice that of the third term” Line 160: “sum their envelope” instead of “sum the envelope of them” The experimental set-up description lacks of details. What kind of EOM is used? What device is used for the 90° hybrid optical coupler? And how are implemented the different pi/2 phase shifts? Details on the set-up (or more simply the device, if the authors used a commercial device) should be added. Would it make sense to replace the EO amplitude modulation (Eq. (2)) by a pure phase modulation? A short comment on this might be added in the paper (optional). Line 88, 89: delta_t is a time delay, and theta denotes a phase difference. Please make the correction Line 157, I assume that it is the amplitude of the “third term” instead of “first term” that varies randomly Line 159: an alternative method is proposed for signal processing, but not fully justified. Some details are required in order to explain why in principle this method enables to increase the detection probability, as it is not intuitive. The authors should at least give an insight on why better results are to be expected, or a reference where this alternative method is detailed. Line 177: the authors indicate that the time zero pulse due to crosstalk is “still unstable”. Some precision should be added. Is it phase instability or amplitude instability? Could the authors give an order of magnitude of the uncertainty on the first pulse amplitude? The authors should also quantify how this uncertainty contributes to the uncertainty on the distance measurement. Fig. 6. The authors use the term “variance” to refer to the noise on the ranging measurement; they should replace it with the term “standard deviation”.

Round 2
Reviewer 1 Report
The significant digits should be checked, for example the values of distance and standard variance in Fig.6. what is the accuracy of the system? The description “ the accuracy increased from 0.45 m to 0.26 m” is not enough.
